# Pan-Genome miRNomics in *Brachypodium*

**DOI:** 10.3390/plants10050991

**Published:** 2021-05-16

**Authors:** Tugdem Muslu, Sezgi Biyiklioglu-Kaya, Bala Ani Akpinar, Meral Yuce, Hikmet Budak

**Affiliations:** 1Faculty of Engineering and Natural Sciences, Molecular Biology, Genetics and Bioengineering Program, Sabanci University, Istanbul 34956, Turkey; tugdem@sabanciuniv.edu (T.M.); sbiyiklioglu@sabanciuniv.edu (S.B.-K.); 2Montana BioAgriculture, Inc., Missoula, MT 59802, USA; aniakpinar@gmail.com; 3Sabanci University SUNUM Nanotechnology Research and Application Centre, Sabanci University, Istanbul 34956, Turkey; meralyuce@sabanciuniv.edu

**Keywords:** microRNA, *Brachypodium*, pan-genome

## Abstract

Pan-genomes are efficient tools for the identification of conserved and varying genomic sequences within lineages of a species. Investigating genetic variations might lead to the discovery of genes present in a subset of lineages, which might contribute into beneficial agronomic traits such as stress resistance or yield. The content of varying genomic regions in the pan-genome could include protein-coding genes as well as microRNA(miRNAs), small non-coding RNAs playing key roles in the regulation of gene expression. In this study, we performed in silico miRNA identification from the genomic sequences of 54 lineages of *Brachypodium distachyon*, aiming to explore varying miRNA contents and their functional interactions. A total of 115 miRNA families were identified in 54 lineages, 56 of which were found to be present in all lineages. The miRNA families were classified based on their conservation among lineages and potential mRNA targets were identified. Obtaining information about regulatory mechanisms stemming from these miRNAs offers strong potential to provide a better insight into the complex traits that were potentially present in some lineages. Future work could lead us to introduce these traits to different lineages or other economically important plant species in order to promote their survival in different environmental conditions.

## 1. Introduction

Genome sequencing has come a long way since the Sanger method was developed in 1975 [1] and reached a major milestone in 2005 with the emergence of next generation sequencing (NGS), which enabled the sequencing of huge quantities of DNA data faster and cheaper than ever before [2,3]. The advent of high throughput NGS technology over the past 15 years has not only tremendously increased the number of de novo genome assemblies, but also revolutionized the way we perceive genomic analysis [4]. The availability of whole genome sequences opened the possibility of studying comparative genomics, which revealed conserved and diverged genomic features of different organisms.

Life on Earth evolves and adapts to different environmental conditions through genetic heterogeneity of organisms driven by mutations. Genes with crucial functions among all organisms are conserved during evolution while other genes differentiate species or cause intra-species variations [5,6]. Apart from gene level variations, single nucleotide polymorphisms (SNPs) contribute equally to intra-specific genetic variations [7]. Even though characterizations of such variants are mostly carried out using a single reference genome-based approach, highly polymorphic regions, presence/absence variations (PAVs) and copy number variations (CNVs) which diverge from the reference genome are unavoidably lost [8,9]. It has been previously shown that 20% of the genes in an agronomically important crop, *Brassica oleracea*, are affected by PAVs; therefore, capturing non-reference sequences is vital in studying genetic diversity within species [10]. A more detailed technique to capture intra-specific variations was pioneered by Tettelin et al. (2005) in which the term “pan-genome” was used for the first time [11].

Pan-genome can be defined as the whole genomic repertoire or union of entire genes of all individuals belonging to a single species or a phylogenetic clade. This repertoire is more extensive than any of a single strain of the species and can be categorized into two: (1) the core genome, and (2) the dispensable genome [5]. The core genome consists of genes present in all individuals of the clade and those genes have been shown to be mainly responsible for basic biological activities and the phenotype. On the other hand, cloud genes of the dispensable genome are present in only a few strains and are not essential for main biological functions, but they contribute to genetic diversity in a beneficial way [5,11].

With an ever-increasing global food demand, attaining higher-yielding and more resilient crops, which is the ultimate aim of agricultural studies, has become greatly dependent upon a good understanding of molecular mechanisms. Plants are exposed to various biotic and abiotic stresses because of their sessile nature; hence, consequently they develop various response mechanisms to cope with stress conditions [12]. The construction of plant pan-genomes explores the genetic diversity and enables characterization of gene variants. Model organisms such as *Brachypodium distachyon* have a great place in genomic studies, revealing complex networks controlling molecular mechanisms like stress response [13]. Recently the pan-genome of *Brachypodium distachyon* was constructed from de novo genome assemblies of 54 lineages. The pan-genome was shown to include twice as many genes as the genome of an individual and the genes forming the dispensable genome were shown to function in selective advantages such as adaptation, defense and development [8].

Besides protein-coding genes, which comprise only a small fraction of the genome in most species, the phenotypic landscape of organisms is also heavily influenced by non-coding sequences. Many studies have shown that plant ncRNAs are differentially expressed under unfavorable environmental conditions for adaptation and enhancing growth and development [14]. MicroRNAs (miRNAs) are 18–24 nucleotide long, endogenous non-coding RNAs which negatively regulate gene expression and control key developmental plasticity, disease resistance and stress response mechanisms [15]. In general, miRNAs regulate the expression of numerous genes through translational repression or transcript cleavage [16]. Because of their regulatory role in gene expression, interest has increased in identifying miRNAs and their functions in plants over the last decade. However, despite the vast amount of plant genome miRNA identifications and characterizations, to the best of our knowledge, miRNAs have not yet been explored at the pan-genome level yet. In this study, we explored the miRNA contents of the de novo genome assemblies of 54 *Brachypodium* lineages [8] to gain insight into the miRNA evolution of *Brachypodium* through the identification of conserved and rare miRNA families, the variation of microRNA abundance and function, and to uncover the molecular basis of agronomic traits that are common or specific to some of the lineages.

## 2. Results

### 2.1. miRNA Identification Resulted in an Average of 90 miRNA Families in Each Lineage

In order to explore the conservation and diversity of miRNA families across the *Brachypodium* pan-genome, de novo genome assemblies of 54 lineages, published in an earlier study [8], were used for miRNA identification. Homology-based in silico miRNA identification identified a total of 115 miRNA families processed from 168,657 miRNA precursors (pre-miRNAs) across all 54 lineages, representing the pan-genome for miRNAs in *Brachypodium* with an average miRNA family count of 90 miRNA families per lineage. Mature miRNA sequences and premiRNA sequences identified in each lineage are shown in Appendix A. Bd1-1 lineage was found to be the lineage with lowest miRNA family number, 83, and the highest number of miRNA family identified in a lineage was 93 (Figure 1).

Of the 115 miRNA families, we identified 11 miRNA families, which have not been previously reported as Brachypodium miRNAs to the best of our knowledge, namely miR1130, miR165, miR2873, miR5161, miR5522, miR5566, miR5568, miR6197, miR6224, miR8155 and miR9783, but have been reported in related grasses [17,18,19,20,21,22,23].

Familywise, we observed that the majority of miRNA families were identified in all or most of the lineages and only a few families were found in progressively fewer lineages (Figure 1). Specifically, 56 of the 115 miRNA families were identified in all 54 lineages and 67 miRNA families were present in more than 98% of them. Thirteen miRNA families were found in only one or two lineages, although it should be noted that the presence or absence of miRNA families within our approach depended on the completeness of the genome assemblies and the accuracy of our predictions, as well as computational approaches. Thus, miRNA families that were identified in only a few lineages can still be present in other lineages with less complete genome sequences.

To further explore clues into what roles highly or rarely conserved miRNAs play, we grouped the miRNA families into four groups based on their level of conservation in 54 lineages (Figure 1). The first group was the “rare” miRNAs group, which consisted of 20 miRNA families identified in fewer than 10 lineages. Thirteen of these miRNA families were present in only one or two lineages and may be considered as lineage specific miRNAs, keeping in mind the limitations of sequencing and our predictions. Ten miRNA families identified in more than 10 but fewer than 45 lineages were grouped under “moderately conserved” miRNAs. The miRNA families conserved in more than 80% of lineages were classified under “highly conserved” miRNA families group. This group consisted of 18 miRNA families which were identified in more than 45 but fewer than 53 lineages. The fourth group, which had the highest number of miRNA families, was the “common” miRNAs group and included miRNA families identified as present in 53 or all 54 lineages.

A closer look at the 168,657 pre-miRNAs sequences, giving rise to the miRNAs, indicated substantial variations in the number of precursor sequences per miRNA family (Figure 2). A few miRNA families had extremely high numbers of precursors across all lineages, such as miR5174 (44,403), miR5181 (43,651), miR5049 (22,340) and miR5175 (20,694). Within each lineage, the pre-miRNA distributions were also similar (Appendix A). Comparison of the precursors against known repeats from Poaceae species revealed that 48 miRNA families were characterized by pre-miRNA sequences containing repetitive elements by more than 50% of their lengths (repetitive pre-miRNAs hereafter). In most cases, repetitive pre-miRNAs represented all or none of the precursors within a miRNA family (Figure 2, Appendix A). Among the non-repetitive pre-miRNAs, we also explored sequence conservation within miRNA families by aligning all precursors from all lineages. We observed that while some miRNA families had precursors aligning perfectly or near-perfectly across different lineages, such as miR394, some other miRNA families, such as miR156, had precursors aligning around the mature miRNA and miRNA* regions, but otherwise contained large gaps within the alignment. In general, pre-miRNAs belonging to the miRNA families that were identified in specific phyla, such as miR7745 and miR7763 (Table 1), seem to retain considerable sequence homology across their entire lengths in different lineages. In contrast, miRNA families widely found across the plant kingdom, such as miR156, miR160, miR166 and miR395, appeared to have accumulated considerable sequence variation. Additionally, these miRNA families had relatively high numbers of precursor sequences, which, at least for non-repetitive pre-miRNAs, may indicate the presence of multiple copies within the *Brachypodium* genomes. Representative pre-miRNA alignments from rare, moderately conserved, highly conserved and common miRNA families are given in Appendix A.

### 2.2. Pairwise Comparison of Common microRNA Families between Lineages Indicates High Conservation

To explore the extent of conservation of miRNA families between lineages, we identified the number of common miRNA families between each pair of lineages. To account for differences in the total numbers of miRNA families predicted in each lineage, we described the extent of miRNA conservation in a lineage as the ratio of miRNA families shared with a second lineage over the total number of the families identified for that lineage (Figure 3). For instance, 96.7% of miRNA families in Arn1 lineage was shared with Mon3 lineage, making the extent of miRNA family conservation 0.967 in Arn1 with Mon3. Notably, this approach introduces a directionality; miRNA family conservation in Mon3 with Arn1 is 0.989. This analysis showed that miRNA families are shared by more than 80% in each pairwise comparison. Additionally, some lineages appeared to have more miRNA families in common than others. For example, of the miRNA families identified in Mon3, a higher number of families are shared with Arn1 than with ABR8 (Figure 3).

### 2.3. mRNA Target Analysis Determined the Potential Biological Processes Targeted by Each miRNA Groups

Potential mRNA targets of the identified mature miRNAs were predicted using psRNATarget tool [24] among the coding sequences of each lineage [8]. Maximum UPE and expectation parameters were adjusted to allow high confidence targets to be retained. In total, mature miRNA sequences from 111 families retrieved 109,438 predicted targets in all 54 lineages (Appendix A). No mRNA targets for the four miRNA families, miR1139, miR5054, miR7709 and miR7748, were obtained. Two of those miRNAs, miR7709 and miR7748, were found to be lineage specific miRNAs which have not been previously reported. While the miRNA families and their respective targets within each lineage give clues into the regulatory networks contributing to the overall characteristics of those lineages, from the pan-genome perspective, we sought to explore the regulatory roles of highly-to-rarely conserved miRNA families across all lineages. To provide a global view, we combined all predicted targets of all miRNA families from all lineages within each of the four groups previously described, in order to generate global groups of targets of rare, moderately conserved, highly conserved, and common miRNAs. We hypothesized that the predicted targets of common miRNA families would be associated with essential pathways, while the predicted targets of the rare miRNAs could identify dispensable but critical pathways related to species diversity. To eliminate redundancy coming from homologous transcripts of different lineages, we clustered the target transcript sequences based on sequence identity and retained only the longest cluster representative. This resulted in 123 clusters for rare miRNA targets, 115 clusters for moderately conserved miRNA targets, 647 clusters for highly conserved miRNA targets, and 6131 clusters for common miRNA targets from the starting total of 109,438 target transcripts. Gene Ontology (GO) annotations were obtained by comparing representative sequences against all Viridiplantae proteins using Blast2Go [25] (Figure 4). Biosynthetic process, cellular protein modification, and transport and nucleobase-containing compound metabolic processes were found to be the major biological processes putatively targeted by all four groups. It appeared that the regulation of core biological processes, such as biosynthesis and protein modification, potentially carried out by common miRNAs found in most lineages, was also contributed to heavily by moderately conserved and even rare miRNAs. Interestingly, the potential targets of only moderately conserved miRNAs were involved in other important processes such as signal transduction, response to chemicals, reproduction, and endogenous stimulus. Response to stimulus and cell communication processes were targeted only by rare miRNA families.

### 2.4. miRNAs Mostly Target Multiple mRNA Sequences but a Transcript Is Rarely Targeted by Multiple miRNAs

Within each lineage, there were instances of miRNA families targeting multiple coding sequences and, conversely, coding sequences targeted by multiple miRNA families, indicating regulatory networks. For each lineage, we extracted the number of miRNAs having one or multiple predicted targets. Similarly, we also extracted the number of predicted targets targeted by one or multiple miRNA families for each lineage. For miRNAs with multiple predicted targets, we observed that, on average, 21 miRNA families were targeting 2–5 coding sequences across lineages (Figure 5a). The mean number of miRNA families targeting only one coding sequence is five and in rare instances, some miRNA families putatively targeted even hundreds of targets, although we cannot rule out that some of these targets could be false positives. For target sequences, on the other hand, most predicted target sequences appeared to be targeted by only one miRNA family (Figure 5b). Therefore, our observations suggest that while it is fairly common for a miRNA family to target a few different sequences, it is not very common for two different families to act on the same target sequence. The miRNA families putatively acting on different target sequences may signify crosstalk between different pathways. On the other hand, target transcripts predicted to be targeted by different miRNA families may indicate strategies to fine tune target expression in response to specific conditions or needs.

We hypothesized that miRNAs acting together in all or most lineages may point to the most functionally conserved networks. We listed all miRNA teams, >1 miRNA putatively acting on a single transcript in a given lineage, and extracted teams that were conserved in all 54 or 53 lineages. Five miRNA teams, miR5049 and miR5174, miR156 and miR529, miR5183 and miR7777, miR1439 and miR5174, and miR397 and miR164, were predicted to be targeting at least one common transcript together in all 54 lineages. Additionally, 4 miRNA teams were predicted to be targeting the same transcript together in 53 lineages. These were miR5185 and miR2275 (except in Tek-4), miR5185 and miR2118 (except in Bd3-1), miR166 and miR165 (except in BdTR8i), and miR5175 and miR5049 (except in Tek-2). While previous findings suggested functional roles for a few of these miRNA teams, others require further studies to unravel potential pathways linked to these miRNA families seemingly functioning together in several different lineages. It should be noted that near-exact miRNA teams (such as miRxxx and miRyyy in some lineages and miRxxx, miRyyy and miRzzz in others) were missed in our stringent approach where only exact matches were extracted. These miRNA teams may represent highly conserved regulatory networks that are finely controlled by the action of a pair of miRNAs.

## 3. Discussion

Advances in next-generation sequencing have enabled extensive use of high-throughput technologies in many studies and the accessibility of RNA sequences has paved the way for studying other types of RNAs besides mRNA. Such studies have revealed the importance of noncoding RNAs in plant gene expression and stress response mechanisms [14]. MicroRNAs are genome-level regulators of gene expression, and identifying miRNAs and their targets provides an understanding of complex regulatory mechanisms [26]. While the miRNA identification of the *Brachypodium* genome has been the subject of many previous studies, to the best of our knowledge, no comprehensive miRNA identification and target analysis of multiple lineages has been conducted previously.

In this study, we identified miRNA families of 54 *Brachypodium* lineages, carried out the target analysis of all mature miRNAs using coding sequences of corresponding genomes, and revealed the potential targets of different miRNA groups. Eighty-five of 115 miRNAs identified in our study were shown to be conserved in more than 80% of 54 lineages. Our predictions included many already-known and experimentally validated *Brachypodium* miRNAs (bdi-miRNAs), such as miR156, miR159, miR160, miR164, miR393, miR528, miR5176, miR5200 and miR5202, some of which are also highly conserved among other monocots [27,28,29]. We identified many well-known plant miRNAs, such as miR156, miR159, miR160, miR393 and miR397, which are conserved between monocots and dicots, and widely found in most flowering plants [30].Consistent with our classification (Table 1), miRNA families common to all or most *Brachypodium* lineages contained families found typically in land plants (Embryophyta), but those that were found in progressively fewer lineages contained families that are so far specific to flowering plants (Magnoliophyta). Additionally, 11 miRNA families, which were previously reported in close relatives but not in *Brachypodium*, were identified [22,23,27]. For instance, researchers previously identified miR5566, miR5568 and miR6224 in sorghum [31]; miR1130, miR9783 and miR6197 in wheat [18,20,32]; and miR2873 in rice [19], but to the best of our knowledge, none of these miRNAs have been identified in *Brachypodium*, including in recent studies [22,23]. These suggest that our approach allows robust identification of bona fide miRNA sequences using next-generation sequencing data. Notably, among the 11 miRNA families not previously reported in *Brachypodium*, three families, miR2873, miR5161 and miR5566 were specific to one or two lineages, a fact which would likely be missed in conventional approaches or lineages frequently used in studies, including the reference species. Another six families, miR165, miR1130, miR5522, miR5568, miR6197 and miR9783, were highly conserved and common to 53 lineages or more, which may point to the importance of high-throughput data in identifying RNA species, such as miRNAs, which may have developmental-stage-specific or spatio-temporal expression patterns that can be missed in small-scale studies. Overall, 91 miRNA families were identified in *Brachypodium* reference genome Bd21, and the other 24 miRNA families identified in this study would likely be missed without the comprehensive pan-genome approach.

An inspection of the pre-miRNA sequence revealed that repetitive sequences made up the majority, if not all, of the precursors of some miRNA families, in particular those with extremely high pre-miRNA counts. It is tempting to speculate that these families, namely miR5049, miR5174, miR5175 and miR5181, which appear to be specific to monocot species (Table 1) might have been generated and proliferated subsequent to a Transposable-Element (TE)-capture after the divergence of monocot species. Alternatively, these families may in fact be mis-annotated siRNAs [33]. Additionally, we observed that pre-miRNA sequences for some families were identical or extremely similar in different lineages, such as miR7745 and miR7763. These mostly corresponded to relatively less conserved miRNA families (Table 1). On the other hand, other families, mostly from the common miRNA group, exhibited considerable variation among precursors from different lineages. Pre-miRNA secondary structures are critical to the correct processing of miRNAs, which mandates sequence constraints on primary sequences which are not yet fully realized in plants [34]. However, it has also been observed that relatively young non-conserved miRNAs typically have few copies in the genomes and retain extensive homology to their targets beyond the mature miRNA subsequence [34,35]. Such miRNAs might then be expected to retain sequence conservation toward the entire length of the respective pre-miRNA sequences. Consistently, we observed such conservation mostly in rarely or moderately conserved miRNA families, which appeared to have limited numbers of precursors in each genome. Conversely, evolution might have more time to act on the precursor sequences of miRNA families found in virtually all plants, usually with multiple copies in the genome [35].

Our predictions identified five miRNA teams targeting at least one common transcript in all 54 lineages. One of these teams, miR156 and miR529, has been previously shown to be evolutionarily related [36,37] and, parallel to these studies, our findings showed that they target SQUAMOSA promoter binding protein (SBP)-box gene family-related transcripts in all 54 lineages. These transcripts encode plant-specific transcription factors and are involved in plant growth and development [38]. Another miRNA team, miR165 and miR166, was determined to target at least one common transcript in 53 lineages, and this finding is also consistent with literature. The miR165/166 family was shown to be present in many plants suggesting its regulatory circuit has been conserved since the last common ancestor of vascular plants [39]. Wojcik et al., (2017) [40] previously showed that miR165/166 regulates the developmental plasticity of somatic cells in vivo, effecting auxin biosynthesis, and is involved in key stress response mechanisms in *Arabidopsis*. We have found 12-oxophytodienoate reductase as a common target of miR165 and miR166, and this protein was previously described in *Arabidopsis* as a stress regulated protein which is involved in jasmonic acid biosynthesis [41].

*Brachypodium* has a high level of genetic variation in its subspecies as a consequence of its allopolyploidization and self-fertilization [42]. Significant differences in population phenotypes, even in a small geographic range, have been observed in previous studies [8,43]. Bd1_1 cultivar was found to have the fewest miRNA families, 83, of all the cultivars in our study. Bd1_1 cultivar from Turkey is a divergent cultivar, based on SSR markers and a late flowering phenotype, which makes this cultivar phenotypically distant from most of the lineages [44]. Of all 54 lineages, Bd1_1 is the only cultivar in which miR7772 was identified.

Based on single nucleotide polymorphisms (SNPs), Gordon et al. [8] deduced phylogenetic relationships among the 54 *Brachypodium* lineages. In terms of miRNA family conservation among given lineages, our observations are in line with these relationships. For instance, BdTR3c is most related to the Koz1 lineage in terms of SNPs, and we identified the same 93 miRNA families in both lineages. Moreover, Arn1 and Ron2 lineages, which have 92 and 90 miRNA families, respectively, have 84 miRNA families in common based on our prediction; supporting this were findings in Gordon et al. [8] that they are distant in terms of SNPs. However, despite the ABR9 and Bd1_1 lineages having been shown to be closely related in terms of SNPs, we identified 78 common miRNA families, which is lower than the average common miRNA family number of 85.5. This difference may be interpreted as the result of a low miRNA family number identified from the Bd1_1 cultivar. We identified miR5200, a conserved *Brachypodium* miRNA which is known to play a role in the regulation of FLOWERING LOCUS T (FT) [45]. Target prediction analyses showed FT was targeted only by miR5200 in all lineages. Mon3 and Arn1 lineages were predicted to have an extra locus than other lineages targeted by miR5200. BdTR7a, BdTR8i, Bd1-1, Bd29-1, Tek2 and Tek-4 were identified to have fewer miR5200 pre-miRNA sequences than other lineages. All six lineages were classified into the extremely delayed flowering phenotype (EDF+) clade in Gordon et al., 2017 [8]. Two other lineages of the same clade, Arn1 and Mon3 lineages, both from Spain, were shown to be closely related based on high confidence SNPs and both have earlier flowering than other lineages of the (EDF+) clade [8]. We identified 89 common miRNAs between these two lineages and also a lineage specific miRNA, miR5161, identified only in Arn1 and Mon3 lineages, which may be further studied in the future to identify its role in the attributed phenotype.

We classified the miRNA families identified in our study into four groups based on their conservation among lineages: 20 miRNA families as rare miRNA families, 10 moderately conserved miRNA families, 18 highly conserved miRNA families, and 67 common miRNA families. Based on this classification, we observed that the majority of miRNA families identified in our prediction were conserved among lineages. A psRNATarget analysis to identify potential mRNA targets of the miRNA families resulted in a high number of target sequences for the 54 lineages. No target sequences were identified for miR1139, miR5054, miR7709 and miR7748 families, which may be a result of the criteria we applied during target identification analysis to eliminate low confidence target sequences.

Target sequences obtained from the psRNATarget analysis were clustered based on at least 90% similarity and the total number of targets with representative sequences was decreased to 7016 for four groups. Gene Ontology enrichment revealed the biological functions in which the four groups, rare, moderately conserved, highly conserved and common miRNAs, were mainly involved. Moderately conserved miRNAs were shown to regulate biological functions, such as response to chemicals and stimulus and signal transduction, which are all key processes in plants’ response mechanisms. Rare miRNA families were found to be involved in cell communication and response to stimulus processes. Plants sense their environments and respond to various biotic and abiotic stresses by altering molecular processes, such as signal transduction [46]. Both rare and moderately conserved miRNA families were found to be involved in regulation of various mechanisms which can contribute to species diversity. However, potential targets of rare miRNA families should be analyzed with caution as low genome completeness, sequencing errors and miRNA identification criteria may result in the presence or absence of some miRNA families in lineages. Hence, moderately conserved miRNA families may provide more accurate insight about lineage differentiation in terms of miRNA evolutions.

## 4. Materials and Methods

### 4.1. Datasets Used in This Study

De novo genome assemblies of 54 *Brachypodium distachyon* lineages were obtained from the study by Gordon et al. [8]. The coding sequences and annotations of rice orthologs of 54 *Brachypodium distachyon* lineages were also retrieved from the public repository of the same study [47].

### 4.2. miRNA Identification

miRNA identification from genomic sequences was performed for each genome separately. A reference set of mature miRNAs was collected by taking all Viridiplantae mature miRNA sequences from miRBase (v21, June 2014) [29]. A two-step homology-based in silico approach was followed as previously described [48,49,50,51]. In brief, using in-house SUmirFind script, candidate miRNA sequences with at most 1 mismatch to the reference miRNAs were searched in the genome. Another in-house script, SUmirFold, was used to generate potential miRNA precursor sequences by extracting and folding them using UNAFold v3.8 algorithm [52], and evaluating them for the presence of known pre-miRNA fold characteristics. Potential pre-miRNA sequences that passed the previous evaluation were tested for additional criteria: (1) no mismatches were allowed at Dicer cut sites, (2) no multi-branched loops were allowed in the hairpin containing the mature miRNA sequence, (3) mature miRNA sequence could not be located at the head portion of the hairpin, and (4) no more than four and six mismatches were allowed in miRNA and miRNA*, respectively, using SUmirScreen to eliminate false-positives. Our two step homology-based pipeline was developed, improved and described in detail, including in-house scripts, in earlier publications [53,54].

### 4.3. Potential mRNA Target Analysis of Identified miRNAs

Potential mRNA targets of the miRNAs were identified using the online web-tool psRNATarget, with user-defined query and target options (http://plantgrn.noble.org/psRNATarget/, accessed on 7 January 2021) [55]. The psRNATarget was run with default parameters, except two: (1) Maximum UPE, the binding energy between miRNA and its target where lower values indicate stronger miRNA-miRNA target interactions, was set to 25, and (2) Expectation, which, similar to e-value in blast, indicates the significance of miRNA-miRNA target pairs, was limited to a maximum of 3. For each lineage, targets were searched among annotated CDS sequences from 54 lineages of *Brachypodium distachyon* [8]. Rice annotations of CDS of each lineage obtained from Gordon et al. 2017 were also used for target prediction of some miRNAs [8].

Clustering of target transcripts were done based on 90% sequence similarity using CD-HIT-EST tool [56]. Representative sequences from each cluster were retained and representative target transcripts within each group were compared to Viridiplantae proteins using blastx tool on a local server (e-value of 1 × 10^6^ and a maximum target of 1). Gene Ontology (GO) annotations for biological process were then obtained using Blast2GO software [25] following mapping, annotation and GO slim steps for plants. Other computational analyses were carried out with in-house Python 3 scripts.

### 4.4. Construction of the Heatmap

The heatmap was drawn on Heatmapper Pairwise Comparison (www.heatmapper.ca/pairwise, accessed on 29 January 2021) [57]. The data for the heatmap was generated using custom Python 3 scripts. For each pairwise comparison, the extent of conservation was defined and calculated as: the extent of miRNA conservation of lineage 1 with lineage 2 = common miRNAs (lineage1, lineage 2)/all miRNAs (lineage 1). From this perspective, the calculation differs for (lineage1, lineage2) and (lineage2, lineage1) comparisons.

### 4.5. Identification of Repetitive Elements and Conservation among Precursor miRNA Sequences

A query of all pre-miRNA sequences of identified miRNA families were searched against the library of Poaceae repetitive elements (MIPS-REdatPoaceae version 9.3) downloaded from ftpmips.helmholtz-muenchen.de/plants/REdat using RepeatMasker tool version 4 with default parameters (http://www.repeatmasker.org/, accessed on 20 February 2021) [58]. Pre-miRNA sequences that contain repetitive elements by more than half of their lengths are considered as ‘repetitive’.

Alignment of precursor miRNA sequences of each miRNA family was done using MAFFT-auto alignment tool with default parameters (https://mafft.cbrc.jp/, accessed on 20 February 2021) [59]. Aligned sequences were analyzed for the conservation of pre-miRNA sequences.

## Figures and Tables

**Figure 1 plants-10-00991-f001:**
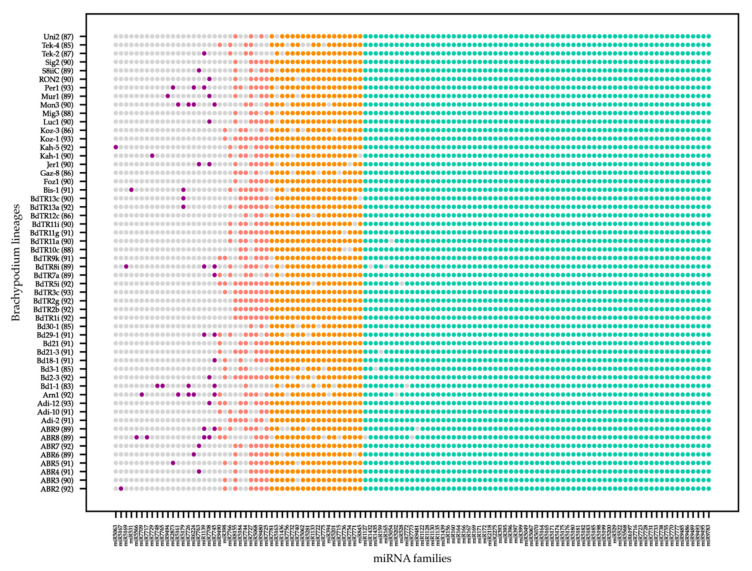
The distribution of miRNA families across lineages. The *x*-axis shows 115 miRNA families identified in 54 *Brachypodium* lineages, shown in *y*-axis. Conservation of miRNA families among lineages increased along the *x*-axis and miRNA families were classified into four groups based on this conservation: Rare miRNA families (purple), moderately conserved miRNA families (pink), highly conserved miRNA families (orange) and common miRNA families (green). The total number of miRNA families identified in each lineage is also shown in parenthesis next to lineage names.

**Figure 2 plants-10-00991-f002:**
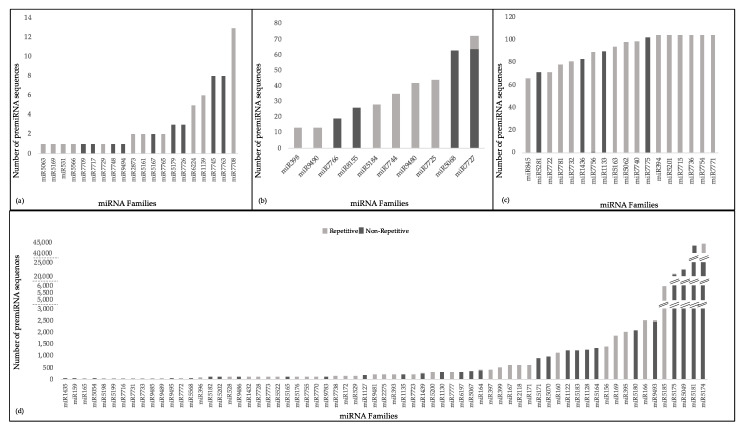
The number of precursor miRNA (pre-miRNA) sequences identified for each miRNA family in (**a**) rare miRNA family group, (**b**) moderately conserved miRNA families group, (**c**) highly conserved miRNA families group, and (**d**) common miRNA families group are shown. Each bar indicates both repetitive pre-miRNAs, defined as containing repetitive elements for more than half of their lengths, with the light shade of gray, and non-repetitive pre-miRNAs with the dark shade of gray.

**Figure 3 plants-10-00991-f003:**
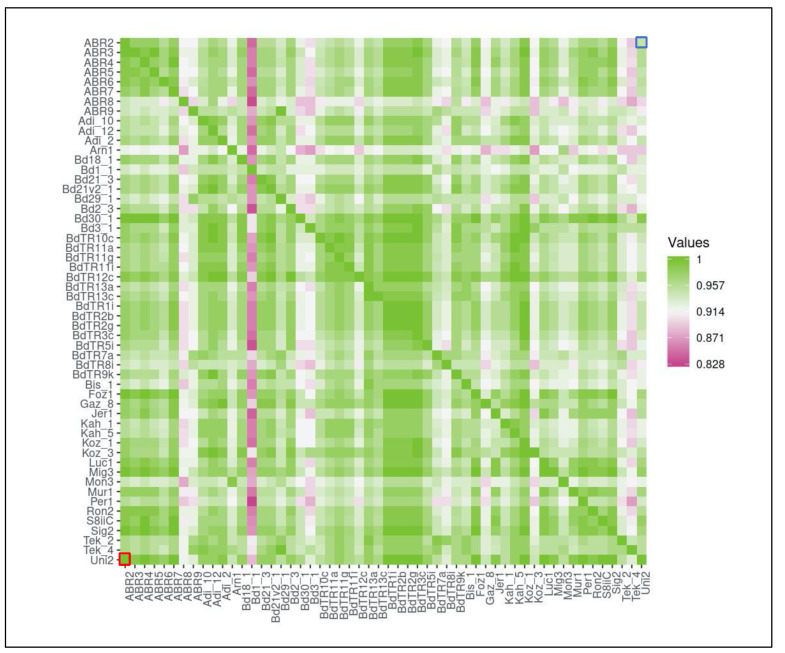
Heatmap displaying the miRNA family commonality between lineages. Darker green cells represent higher numbers of common miRNAs shared between two lineages while darker purple represents fewer numbers of common miRNA families. The heatmap is not symmetrical; it should be read from y > x. For example, the extent of conservation between ABR2 and Uni2, defined as the ratio of common miRNA families between ABR2 and Uni2 over the number of total miRNA families in ABR2 by our approach, is given on the top right corner (indicated by a blue square), while the conservation between Uni2 and ABR2, the ratio over the total number of families in Uni2, is given on the bottom left (red square).

**Figure 4 plants-10-00991-f004:**
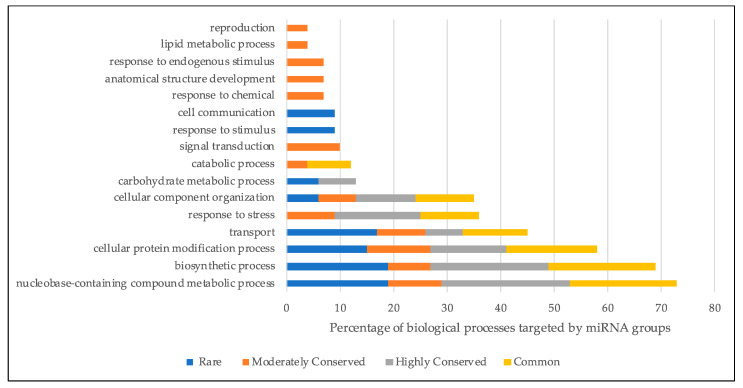
Gene Ontology analysis for biological processes for rare, moderately conserved, highly conserved and common miRNA family groups targets, combined and clustered. Predicted targets from five biological processes appear to be collectively targeted by all four miRNA groups.

**Figure 5 plants-10-00991-f005:**
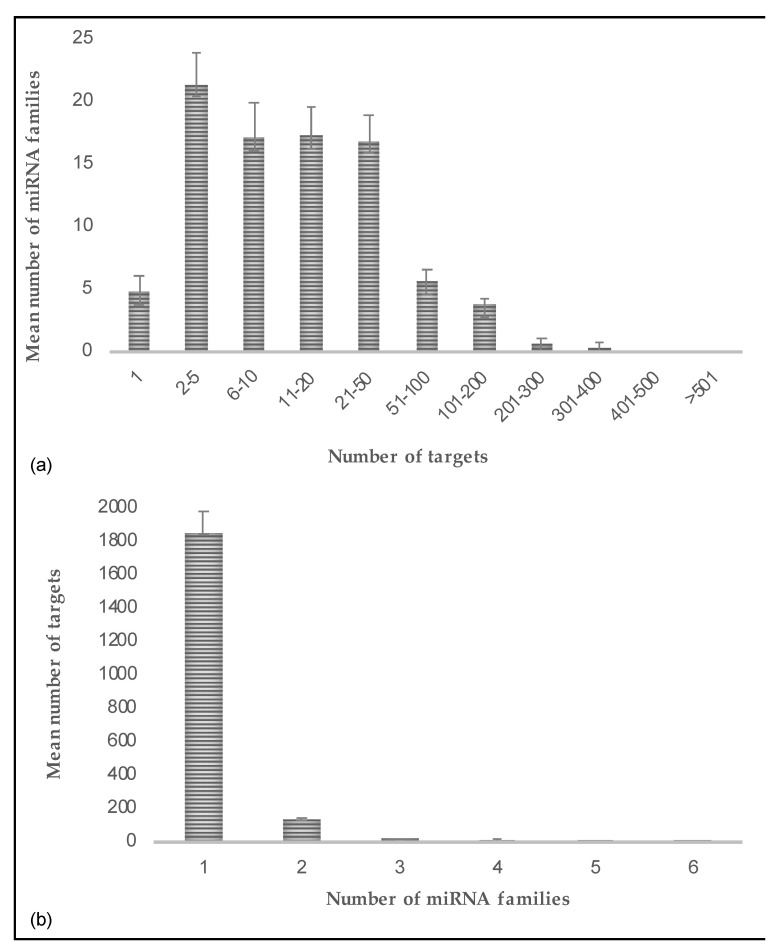
(**a**) The number of miRNA families having multiple predicted targets were averaged across lineages. For simplicity, targets were considered within 11 bins of certain sizes as given. Most miRNA families seemed to target a minimum of 2 and maximum of 50 coding sequences, whereas only a few miRNA families targeted only one target or had more than 50 predicted targets. (**b**) The number of predicted targets of only one or multiple miRNA families were averaged across lineages. In contrast to miRNAs targeting multiple targets, each target appeared to be predominantly targeted by one miRNA family.

**Table 1 plants-10-00991-t001:** All miRNA families identified from de novo genome assemblies of 54 *Brachypodium* lineages are classified based on their presence in multiple lineages. Phylogenetic inference based on miRBase phyla data for each miRNA family is given.

		Magnoliophyta	Coniferophyta	Embryophyta
Rare	miR1139, miR2873, miR5063, miR5161, miR5167, miR5169, miR5179, miR531, miR5566, miR6224, miR7708, miR7709, miR7717, miR7726, miR7729, miR7745, miR7748, miR7763, miR7765, miR9494	√ (m)		
Mod. Cons.	miR398	√	√	
	miR5068, miR5184, miR7725, miR7727, miR7744, miR7766, miR9480, miR9490	√ (m)		
	miR8155	√ (e)		
Highly Cons.	miR394	√		
	miR1133, miR1436, miR5062, miR5163, miR5201, miR7715, miR7722, miR7732, miR7736, miR7740, miR7754, miR7756, miR7771, miR7775, miR7781	√ (m)		
	miR5281, miR845	√ *		
Common	miR1122, miR1127, miR1128, miR1130, miR1135, miR1432, miR1435, miR1439, miR5049, miR5054, miR5067, miR5070, miR5164, miR5165, miR5171, miR5174, miR5175, miR5176, miR5180, miR5181, miR5182, miR5183, miR5185, miR5198, miR5199, miR5200, miR5202, miR528, miR5522, miR5568, miR6197, miR7716, miR7723, miR7728, miR7731, miR7733, miR7738, miR7755, miR7770, miR7772, miR7773, miR7777, miR9481, miR9485, miR9486, miR9489, miR9493, miR9495, miR9783	√ (m)		
	miR156, miR159, miR160, miR166, miR171, miR395, miR396	√	√	√
	miR164, miR169, miR397	√	√	
	miR165	√ (e)		
	miR167	√		√
	miR172, miR393	√		
	miR2118, miR2275, miR399	√ *		

*: All monocotyledons and eudicotyledons, except *Amborella trichopoda*; (m): Only in monocotyledons; (e): Only in eudicotyledons.

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
