# Peer review of "Pan-Genome miRNomics in Brachypodium"

_plants, 2021, doi:10.3390/plants10050991_

Round 1

Reviewer 1 Report

In this manuscript, Muslu and colleagues investigated miRNAs encoded in 54 different lineages of Brachypodium distachyon. The authors found an average of 90 miRNA families in each lineage, with some interesting variation across the different lineages. The authors also used in silico tools to predict mRNA targets of the rare, moderately conserved, highly conserved and common miRNAs in the Brachypodium pan-genome sample. Much of these conclusions remain speculative, given that Parallel Analysis of RNA Ends (PARE) was not used to validate the putative targeting relationships. But this next step could be conducted in a follow-up study of the pan-genome lineages.

The authors should avoid highly speculative statements like "Additionally, 4 miRNA teams, targeting the same transcript together in a given lineage, hence working together, were found in 53 lineages."(Page 10, line 237). The authors did not test whether any miRNAs are effectively targeting transcripts in the Brachypodium lineages. That would require additional data for target cleavage using PARE or an equivalent NGS approach. The language should be that the miRNAs are "predicted to target the same transcript". This is all that the bioinformatic analysis using psRNATarget can reveal.

I would encourage the authors to cite two early studies that identified highly conserved miRNAs, notably assisted by computational analysis of plant genomes:

- Jones-Rhoades & Bartel (2004). Computational identification of plant microRNAs and their targets, including a stress-induced miRNA. Mol Cell 14(6):787-99. doi:10.1016/j.molcel.2004.05.027.

- Axtell & Bartel (2005). Antiquity of microRNAs and their targets in land plants. Plant Cell 17(6):1658-73. doi:10.1105/tpc.105.032185.

The authors identified 11 miRNA families that may not have been previously reported in Brachypodium. It would be essential for the authors to cite, analyse and discuss recent studies on miRNAs in Brachypodium to confirm this conclusion. Two such studies are missing from the reference list:

- Franke et al. (2018). Analysis of Brachypodium miRNA targets: evidence for diverse control during stress and conservation in bioenergy crops. BMC Genomics 19(1):547. doi: 10.1186/s12864-018-4911-7.

- Jeong et al. (2013). Parallel analysis of RNA ends enhances global investigation of microRNAs and target RNAs of Brachypodium distachyon. Genome Biol. 14(12):R145. doi: 10.1186/gb-2013-14-12-r145.

On balance, this manuscript could be of interest to the field of miRNA evolution and function. Researchers who study Brachypodium non-coding RNAs and gene regulation based on the Brachypodium pan-genome (Gordon et al. 2017 Nat. Comm.) will benefit from improved miRNA annotations, specifically obtained from those lineages.

However, the bar charts in Fig. 1 and Fig. 2 are not well-designed. They fill copious space on the page but do not convey the analyses of miRNA families succinctly. I recommend that the authors take time to improve their figure quality, to properly label axes, to include missing information in legends and to convert the bar charts into a more distilled format, such as line or x-y scatter plots. If these revisions are made, then I could recommend publication of the manuscript in Plants.

Minor issues:

1) There are a variety of English usage errors in the manuscript. I recommend that the authors have an English native speaker read through and correct these issues: e.g., Page 1, line 43: "Previously shown that 20% of an agronomically important crop, Brassica oleracea, genes are affected by PAVs ..." --> "It was previously shown that 20% of genes in an agronomically important crop, Brassica oleracea, are affected by PAVs ..."

2) Page 1, line 30 -- The first reference about NGS technology (Grada and Weinbrecht 2013) is rather obscure and out of date. Instead, I would recommend citing Mardis ER (2017). DNA sequencing technologies. Nature protocols https://pubmed.ncbi.nlm.nih.gov/28055035/ doi:10.1038/nprot.2016.182

3) Page 5, line 136/Fig. 2 -- The authors state that the "common" miRNA group includes "miRNA families identified to be present in 53 or all 54 lineages." I gather that this represents a range of the bars in Fig. 2 from the middle to the right-hand side. I recommend that the authors use this graphic to mark the range of bars that are in the four identified categories:

  1. rare
  2. moderately conserved
  3. highly conserved
  4. common

This could be accomplished with horizontal lines or brackets that encompass the corresponding lineages.

4) More generally, I recommend that the authors adjust the data display in Fig. 2 to simplify interpretation. The colors in Fig. 2 are completely inscrutable: no information is offered concerning color values in the legend. Whatever display is used to improve this situation, the x and y-axis labels must have larger fonts to allow readers to identify data points corresponding to individual Brachypodium lineages.

5) Page 6, Table 1 -- This table could be used to summarise more information. For example, which of the miRNA families shown are universally conserved in Embryophyta (miR156, miR159/319, miR160, miR166, miR171, and miR395?), and which miRNA families are specific to particular plant taxa? Also, there is a problem with the alphanumerical sorting order. It would be more logical is the lists in each table column started with the lowest miR ID and increased to the highest miR ID.

Author Response

Response to Reviewer 1:

We would like to thank the reviewer for constructive comments and criticism, which we believe helped us improve our manuscript significantly. Below are our responses to the comments.

Thank you,

Hikmet

Comments and Suggestions for Authors

In this manuscript, Muslu and colleagues investigated miRNAs encoded in 54 different lineages of Brachypodium distachyon. The authors found an average of 90 miRNA families in each lineage, with some interesting variation across the different lineages. The authors also used in silico tools to predict mRNA targets of the rare, moderately conserved, highly conserved and common miRNAs in the Brachypodium pan-genome sample. Much of these conclusions remain speculative, given that Parallel Analysis of RNA Ends (PARE) was not used to validate the putative targeting relationships. But this next step could be conducted in a follow-up study of the pan-genome lineages.

The authors should avoid highly speculative statements like "Additionally, 4 miRNA teams, targeting the same transcript together in a given lineage, hence working together, were found in 53 lineages."(Page 10, line 237). The authors did not test whether any miRNAs are effectively targeting transcripts in the Brachypodium lineages. That would require additional data for target cleavage using PARE or an equivalent NGS approach. The language should be that the miRNAs are "predicted to target the same transcript". This is all that the bioinformatic analysis using psRNATarget can reveal.

Our response: We agree with the reviewer that our conclusions are mostly speculative in the absence of experimental validation, even in cases where our observations are consistent with earlier findings. We have revised our manuscript to ensure our statements reflect that our conclusions are based on computational predictions.

I would encourage the authors to cite two early studies that identified highly conserved miRNAs, notably assisted by computational analysis of plant genomes:

- Jones-Rhoades & Bartel (2004). Computational identification of plant microRNAs and their targets, including a stress-induced miRNA. Mol Cell 14(6):787-99. doi:10.1016/j.molcel.2004.05.027.

- Axtell & Bartel (2005). Antiquity of microRNAs and their targets in land plants. Plant Cell 17(6):1658-73. doi:10.1105/tpc.105.032185.

Our response: We cited these important studies of the field in our revised manuscript.

The authors identified 11 miRNA families that may not have been previously reported in Brachypodium. It would be essential for the authors to cite, analyse and discuss recent studies on miRNAs in Brachypodium to confirm this conclusion. Two such studies are missing from the reference list:

- Franke et al. (2018). Analysis of Brachypodium miRNA targets: evidence for diverse control during stress and conservation in bioenergy crops. BMC Genomics 19(1):547. doi: 10.1186/s12864-018-4911-7.

- Jeong et al. (2013). Parallel analysis of RNA ends enhances global investigation of microRNAs and target RNAs of Brachypodium distachyonGenome Biol. 14(12):R145. doi: 10.1186/gb-2013-14-12-r145.

Our response: We have included these studies in our revised manuscript.

On balance, this manuscript could be of interest to the field of miRNA evolution and function. Researchers who study Brachypodium non-coding RNAs and gene regulation based on the Brachypodium pan-genome (Gordon et al. 2017 Nat. Comm.) will benefit from improved miRNA annotations, specifically obtained from those lineages.

However, the bar charts in Fig. 1 and Fig. 2 are not well-designed. They fill copious space on the page but do not convey the analyses of miRNA families succinctly. I recommend that the authors take time to improve their figure quality, to properly label axes, to include missing information in legends and to convert the bar charts into a more distilled format, such as line or x-y scatter plots. If these revisions are made, then I could recommend publication of the manuscript in Plants.

Our response: We agree with the reviewer and would like to thank for the constructive feedback on Figures 1 and 2. Both figures are revised, and their orders are reversed, based on revisions made on Section 2.1. Figure 1 (now included as Figure 2 in the revised manuscript) is simplified; individual lineage information is excluded and overall pre-miRNA counts are summarized as bar graphs. Figure 2 (now Figure 1 in the revised manuscript) is presented basically as a scatter plot/matrix to indicate presence/absence of miRNA families in different lineages. The presence/absence of miRNA families in lineages, along with the number of pre-miRNA sequences for each miRNA family in each lineage, is given as Supplementary Table S1 in the revised manuscript.

Minor issues:

1) There are a variety of English usage errors in the manuscript. I recommend that the authors have an English native speaker read through and correct these issues: e.g., Page 1, line 43: "Previously shown that 20% of an agronomically important crop, Brassica oleracea, genes are affected by PAVs ..." --> "It was previously shown that 20% of genes in an agronomically important crop, Brassica oleracea, are affected by PAVs ..."

Our response: We would like to thank the reviewer for pointing out language mistakes which helped us improve our manuscript. The above-mentioned sentence is corrected. Additionally, the manuscript is revised for the use of the language.

2) Page 1, line 30 -- The first reference about NGS technology (Grada and Weinbrecht 2013) is rather obscure and out of date. Instead, I would recommend citing Mardis ER (2017). DNA sequencing technologies. Nature protocols https://pubmed.ncbi.nlm.nih.gov/28055035/ doi:10.1038/nprot.2016.182

Our response: This reference is cited in the revised manuscript.

3) Page 5, line 136/Fig. 2 -- The authors state that the "common" miRNA group includes "miRNA families identified to be present in 53 or all 54 lineages." I gather that this represents a range of the bars in Fig. 2 from the middle to the right-hand side. I recommend that the authors use this graphic to mark the range of bars that are in the four identified categories:

  1. rare
  2. moderately conserved
  3. highly conserved
  4. common

This could be accomplished with horizontal lines or brackets that encompass the corresponding lineages.

Our response: Based on the reviewer’s feedback on Figure 2, this figure is fully revised. The revised figure indicates these four categories explicitly.

4) More generally, I recommend that the authors adjust the data display in Fig. 2 to simplify interpretation. The colors in Fig. 2 are completely inscrutable: no information is offered concerning color values in the legend. Whatever display is used to improve this situation, the x and y-axis labels must have larger fonts to allow readers to identify data points corresponding to individual Brachypodium lineages.

Our response: We would like to thank the reviewer for the comments and suggestions on Figure 2. We have revised this figure accordingly.

5) Page 6, Table 1 -- This table could be used to summarise more information. For example, which of the miRNA families shown are universally conserved in Embryophyta (miR156, miR159/319, miR160, miR166, miR171, and miR395?), and which miRNA families are specific to particular plant taxa? Also, there is a problem with the alphanumerical sorting order. It would be more logical is the lists in each table column started with the lowest miR ID and increased to the highest miR ID.

Our response: We would like to thank the reviewer for the comments and suggestions on Table 1. The requested changes are made.

Reviewer 2 Report

Authors present a survey of conserved miRNAs families and precursors that are present in the genomes of 54 sequenced Brachypodium lineages. The paper is quite innovative since it is the first survey on intra-species variability of miRNAs sequences in plants.

The major shortcoming of the paper is that authors present only a survey on the variability of the number of miRNAs, on which miRNAs are shared or lineage specific and on their predicted targets without any insight on genomic sequence variability. It would be interesting to know if SNP, INDELs or other structural variations are present in the pre-miRNAs sequences and in their targets, with a potential effect on secondary structure or miRNA-target recognition.

Using data from the pan-genome paper (ref. 7) authors could add information on miRNAs the reside in partially deleted regions of the pan genome (PAVs), or in highly dissimilar regions. Moreover transcriptomic data, available for some of the lineages could be integrated, to have information, for example, on potential miRNA genes coordinates (i.e. primary transcripts) and hence on putative variation on gene regulatory regions.

Other Major revisions:

  1. Authors present some miRNA families with hundreds of precursors, that raise some doubts about their nature: couldn’t be repetitive elements ?

  1. From heatmap (paragraph 2.2) authors identify groups of accessions that are more similar, sharing a higher number of miRNAs, and other accession sharing a very low number of miRNAs (i.e. the pink/violet columns in the heatmap). It would be interesting to discuss in more details these groups of accession in the perspective of the previous papers (refs 7 and 34). In this direction, authors discuss briefly the Bd1_1 lineage, but it would be worth using all the data of the pan genome to discuss more deeply also other linages.

  1. Paragraph 2.4 is not highly informative. Authors identify teams of miRNAs (some of which are well known to work together, as they point out in the discussion) but they do not further analyze which could be their role and significance

4.The discussion on the TDMD  pathway is not sufficiently supported by evidences coming from the paper

  1. Methods could be improved, when describing the last step, using SUmirScreen, that should eliminate false positive.

Minor revisions:

Line 43 check the syntax of the sentence, a verb is missing

Line 130 check the syntax of the sentence, after “taking into account”

Line 292: check the syntax of the sentence

Fig1b : color code is missing in the caption

Fig2 : please, put on the Y axis bigger numbers. Please also mention X axis, as it’s impossible to read

Lines 157-159 : the calculation is not very clear, maybe it’s better to write the formula

Line 179: authors talk about 2 families, and they should specify which ones

Line204 the figure is numbered as figure 2 but should be figure 4

Author Response

Response to Reviewer 2:

We would like to thank the reviewer for constructive comments and criticism, which we believe helped us improve our manuscript significantly. Below are our responses to the comments.

Comments and Suggestions for Authors

Authors present a survey of conserved miRNAs families and precursors that are present in the genomes of 54 sequenced Brachypodium lineages. The paper is quite innovative since it is the first survey on intra-species variability of miRNAs sequences in plants.

The major shortcoming of the paper is that authors present only a survey on the variability of the number of miRNAs, on which miRNAs are shared or lineage specific and on their predicted targets without any insight on genomic sequence variability. It would be interesting to know if SNP, INDELs or other structural variations are present in the pre-miRNAs sequences and in their targets, with a potential effect on secondary structure or miRNA-target recognition.

Our response: We would like to thank the reviewer for the constructive comments and requests. We have aligned pre-miRNAs sequences for each miRNA family from all lineages included in the study. Although it would be very difficult to call SNPs and INDELs in the absence of preferably high-depth small-RNA read data (all identified variations can simply be sequencing/assembly errors as the miRNA identification was done on assembled genome sequences), we did observe extensive sequence conservation or variation in different miRNA families. These are presented in Section 2.1 of the revised manuscript and also included in the Discussion part. We also provided representative alignments from our rare, moderately conserved, highly conserved and common miRNA categories as Supplementary File S1. If requested, we can also provide all alignments.

Using data from the pan-genome paper (ref. 7) authors could add information on miRNAs the reside in partially deleted regions of the pan genome (PAVs), or in highly dissimilar regions. Moreover transcriptomic data, available for some of the lineages could be integrated, to have information, for example, on potential miRNA genes coordinates (i.e. primary transcripts) and hence on putative variation on gene regulatory regions.

Our response: We agree with the reviewer that it would be very exciting to be able to delineate PAVs and indicate which miRNA families come from such highly variable regions. However, even though we mapped our pre-miRNA sequences on the Brachypodium pangenome (https://brachypan.jgi.doe.gov/), the information available to us at the moment is restricted to variations in the genes, not quite on the genome. In their manuscripts, Gordon et al., 2017, also describe genomic PAVs largely in the context of specific comparisons (such as Fig. 1d, e f). Therefore, although we did attempt to describe pre-miRNA PAVs on the pangenome level, at this moment,  we are not able to fulfil this request.

In our manuscript, we have used transcriptome data from each Brachypodium lineage (from Gordon et al. 2017 Nat. Comm.) in target prediction. While we agree that additional transcriptomic data can provide more insight into miRNA genes, we fear that highly tissue-, developmental-stage- and/or environment-specific expression profiles of some miRNA families and differences between miRNA genes and protein-coding genes (and the fact that most miRNA genes are encoded in the intergenic space in plants) would necessitate an experimental design specifically targeted for miRNA genes for the transcriptome data to be fully useful. The transcriptome data we were able to find were mostly targeted at the identification of differentially expressed genes upon specific physiological conditions.

Other Major revisions:

  1. Authors present some miRNA families with hundreds of precursors, that raise some doubts about their nature: couldn’t be repetitive elements?

Our response: We would like to thank the reviewer for this point. Since the overall repetitive content of the Brachypodium genome is relatively small (against other grasses like wheat, barley etc.), we did not include a repeat analysis initially. Upon the request by the reviewer, we compared all pre-miRNA sequences against known Poaceae repeats and indeed found repetitive elements in our precursor sequences. These observations are presented in Section 2.1 and included in the new Figure 2 of the revised manuscript and also in the Discussion part. Summary information for repetitive and non-repetitive pre-miRNAs (defined as containing repeat elements for >50% of their lengths) are also given in Supplementary Table S1.

  1. From heatmap (paragraph 2.2) authors identify groups of accessions that are more similar, sharing a higher number of miRNAs, and other accession sharing a very low number of miRNAs (i.e. the pink/violet columns in the heatmap). It would be interesting to discuss in more details these groups of accession in the perspective of the previous papers (refs 7 and 34). In this direction, authors discuss briefly the Bd1_1 lineage, but it would be worth using all the data of the pan genome to discuss more deeply also other linages.

 Our response: We have expanded our discussions on the heatmap data in the revised manuscript.

  1. Paragraph 2.4 is not highly informative. Authors identify teams of miRNAs (some of which are well known to work together, as they point out in the discussion) but they do not further analyze which could be their role and significance

 Our response: We would like to thank the reviewer for the feedback in Section 2.4. This section is partially revised to reflect potential functions of the miRNA teams might play.

4.The discussion on the TDMD pathway is not sufficiently supported by evidences coming from the paper

 Our response: Upon the reviewer’s feedback, the TDMD pathway part in the discussion is removed.

  1. Methods could be improved, when describing the last step, using SUmirScreen, that should eliminate false positive.

Our response: We would like to thank the reviewer for the comments on the Methods. We noticed a confusion with the description of SUmirScreen. We have corrected this and cited two of our most relevant papers describing our computational miRNA identification approach in detail, including the in-house scripts. We also expanded parts of the Methods section and added new analyses to the revised manuscript.

Minor revisions:

Line 43 check the syntax of the sentence, a verb is missing

Line 130 check the syntax of the sentence, after “taking into account”

Line 292: check the syntax of the sentence

Our response: The requested corrections were made. Additionally, the manuscript is revised by an academic communication specialist from Sabanci University for the use of the language.

Fig1b : color code is missing in the caption

Fig2 : please, put on the Y axis bigger numbers. Please also mention X axis, as it’s impossible to read

Our response: We would like to thank all reviewers for the constructive comments on both Figures 1 and 2. As a result, we completely revised both figures.

Lines 157-159 : the calculation is not very clear, maybe it’s better to write the formula

Our response: The formula is given explicitly in the Methods section.

Line 179: authors talk about 2 families, and they should specify which ones

Our response: Names of the two miRNA families are added to the sentence

Line204 the figure is numbered as figure 2 but should be figure 4

Our response: The figure numbers are corrected.

Reviewer 3 Report

In this study, authors have identified miRNA families of 54 Brachypodium lineages, carried out the target analysis of all mature miRNAs using coding sequences of corresponding genomes and revealed the targets of different miRNA groups. This is a very interesting study of miRNAs families and distribution in different lineages. I believe the information will be very useful for the scientific community working on miRNAs. The expression of some of the miRNAs and their targets could have been validated experimentally, however I understand this is an in silico study. I suggest a revision throughout the manuscript to correct minor mistakes, some of which are listed below:

Line 43 : change to “Previously it was shown that”

Figure 1 – in a) y axis please correct “families”

Correct some small mistakes or missing words such as:

Line 130

Line 134

Line 266

Line 280

Line 284

Line 292

Line 309

Line 335

Table 1 – avoid beginning a sentence with numbers

Author Response

Response to Reviewer 3:

We would like to thank the reviewer for constructive comments and criticism, which we believe helped us improve our manuscript significantly. Below are our responses to the comments.

Comments and Suggestions for Authors

In this study, authors have identified miRNA families of 54 Brachypodium lineages, carried out the target analysis of all mature miRNAs using coding sequences of corresponding genomes and revealed the targets of different miRNA groups. This is a very interesting study of miRNAs families and distribution in different lineages. I believe the information will be very useful for the scientific community working on miRNAs. The expression of some of the miRNAs and their targets could have been validated experimentally, however I understand this is an in silico study. I suggest a revision throughout the manuscript to correct minor mistakes, some of which are listed below:

Our response : We would like to thank the reviewer for the constructive comments and suggestions. We have corrected the small mistakes and improved the manuscript with the help of a Sabanci University academic communication specialist.

Line 43 : change to “Previously it was shown that”

Our response: The sentence is changed as requested.

Figure 1 – in a) y axis please correct “families”

Our response: Based on reviewer comments, Figure 1 is thoroughly revised. In the revised version, the axis label is corrected as requested.

Correct some small mistakes or missing words such as:

Line 130

Line 134

Line 266

Line 280

Line 284

Line 292

Line 309

Line 335

Our response: We would like to thank the reviewer for the suggestions. The requested corrections are made, and the manuscript is edited.

Table 1 – avoid beginning a sentence with numbers

Our response: Based on reviewer comments, Table 1 is revised. Additionally, other parts of the manuscript are also edited based on this comment.

Round 2

Reviewer 2 Report

The new version of the paper has been corrected mainly following the referees' requests. Anyway, authors could not improve the paper significantly, introducing solid analyses on sequence variatons or transcriptomic data. In the present form the paper clearly presents and discuss the data collected from the 54 Brachypodium lineages, but on the whole data in the present form are not sufficient to achieve the scientific level of the journal, since they are only a  description of the number and distribution of miRNA precursors and targets across the lineages.